# Calcium Induces the Cleavage of NopA and Regulates the Expression of Nodulation Genes and Secretion of T3SS Effectors in *Sinorhizobium fredii* NGR234

**DOI:** 10.3390/ijms25063443

**Published:** 2024-03-19

**Authors:** Wonseok Kim, Sebastián Acosta-Jurado, Sunhyung Kim, Hari B. Krishnan

**Affiliations:** 1Division of Plant Science and Technology, University of Missouri, Columbia, MO 65211, USA; wonseokk@missouri.edu (W.K.); jeongs@missouri.edu (S.K.); 2Departamento de Biología Molecular e Ingeniería Bioquímica, Centro Andaluz de Biología del Desarrollo, Universidad Pablo de Olavide/Consejo Superior de Investigaciones Científicas/Junta de Andalucía, 41013 Sevilla, Spain; sacosta@us.es; 3Plant Genetics Research Unit, USDA, Agricultural Research Service, Columbia, MO 65211, USA

**Keywords:** calcium, NopA, *Sinorhizobium fredii* NGR234, symbiosis, T3SS effectors

## Abstract

The type III secretion system (T3SS) is a key factor for the symbiosis between rhizobia and legumes. In this study, we investigated the effect of calcium on the expression and secretion of T3SS effectors (T3Es) in *Sinorhizobium fredii* NGR234, a broad host range rhizobial strain. We performed RNA-Seq analysis of NGR234 grown in the presence of apigenin, calcium, and apigenin plus calcium and compared it with NGR234 grown in the absence of calcium and apigenin. Calcium treatment resulted in a differential expression of 65 genes, most of which are involved in the transport or metabolism of amino acids and carbohydrates. Calcium had a pronounced effect on the transcription of a gene (NGR_b22780) that encodes a putative transmembrane protein, exhibiting a 17-fold change when compared to NGR234 cells grown in the absence of calcium. Calcium upregulated the expression of several sugar transporters, permeases, aminotransferases, and oxidoreductases. Interestingly, calcium downregulated the expression of *nodABC*, genes that are required for the synthesis of nod factors. A gene encoding a putative outer membrane protein (OmpW) implicated in antibiotic resistance and membrane integrity was also repressed by calcium. We also observed that calcium reduced the production of nodulation outer proteins (T3Es), especially NopA, the main subunit of the T3SS pilus. Additionally, calcium mediated the cleavage of NopA into two smaller isoforms, which might affect the secretion of other T3Es and the symbiotic establishment. Our findings suggest that calcium regulates the T3SS at a post-transcriptional level and provides new insights into the role of calcium in rhizobia–legume interactions.

## 1. Introduction

A group of soil proteobacteria, known as rhizobia, are able to establish a mutualistic interaction with leguminous plants. As a result, the plant forms new root organs called nodules where rhizobia are differentiated into bacteroids, a new physiological and morphological state able to fix atmospheric nitrogen to ammonia [1,2]. The success of this process relies on a complex and coordinated molecular signal interchange between both actors, since only compatible rhizobial strains will be able to induce the formation of effective and, consequently, nitrogen-fixing nodules [3].

The molecular dialogue starts with the release of a set of molecules by the plant roots, including flavonoids, phenolic compounds that interact with the bacterial protein regulator NodD [4,5]. This interaction results in the binding of NodD to specific promoter sequences, called *nod* boxes (NBs), and the consequent induction of genes located downstream of these NBs, which are mostly involved in the symbiotic process [6]. Among these induced genes are the *nod* genes, whose function is the production and secretion of the symbiotic signals named nod factors (NF). The perception of the NFs by plant root LysM receptors trigger bacterial infection and nodule organogenesis [7,8,9].

In addition to NF, other bacterial molecular signals are involved in the success of the symbiotic process, such as a set of secreted proteins collectively known as Nops (nodulation outer proteins). These proteins are secreted by a type III secretion system (T3SS) in a NodD-flavonoid dependent manner. NodD activates the expression of TtsI, a transcriptional regulator that recognizes and binds specific promoter sequences, called *tts* boxes, located upstream of T3SS-related genes. Thus, TtsI induces the expression of genes involved in both the T3SS machinery assembly and the production of Nops. These proteins are delivered through the T3SS into host plant cells where they may alter host pathways or suppress plant defense responses. Therefore, Nops are involved in nodulation efficiency and host-range determination [10,11,12,13,14,15].

*Sinorhizobium fredii* NGR234 (hereafter NGR234) is a fast-growing rhizobial strain isolated from *Lablab purpureus* nodules from Papua New Guinea [16]. Unlike other *Sinorhizobium fredii* strains, NGR234 possesses the broadest nodulation host range known so far, nodulating 112 genera including legumes and the non-legume *Parasponia andersonii* [17]. The symbiotic effects caused by defects in the Nops secretion can be beneficial, detrimental, or have no effect, depending on the specific rhizobia/legume couple. Even the recognition of one Nop can completely block the nodulation [13,18,19]. In the case of a NGR234 *rhcN* mutant, unable to secrete any Nops, the symbiotic effects can vary depending on the plant tested, producing an increase, no changes, or a decrease in the nodule number [18]. These changes are not only produced among different plant species but also among different cultivars, for example as occurs in the *G. soja* cultivars CH2, CH3, and CH4 inoculated with *S. fredii* HH103 T3SS mutants, which are able to induce nitrogen-fixing nodules in CH2 (while the wild type is unable) or cause a nodulation impairment in CH3 and CH4 [20]. In the same strain, the T3SS abolition also provokes an increase in the nodulation host range, allowing the appearance of nitrogen-fixing nodules in the plant model legume *Lotus japonicus* Gifu (the wild type only produces ineffective nodules) and switching the infection mode from intercellular infection to a more evolved one by infection thread formation in *L. burttii* [18]. 

In a previous report, it was demonstrated that Nops production was drastically reduced in the extracellular media of flavonoid-induced culture of *S. fredii* USDA257 in the presence of calcium. This effect seems to occur at post-transcriptional level and is calcium specific since other cations such as magnesium or manganese had no effect on Nops production in the same conditions [21]. In this work, we have investigated the effect of calcium on the overall NGR234 gene expression in the absence and presence of the inducer flavonoid apigenin through RNA-seq experiments. Our results show that calcium supplemented at 0.5 mM concentration is able to modulate the expression of a specific set of NGR234 genes, even in the presence of apigenin, since the gene expression profile differs in the presence of both elements. In addition, we have investigated the expression of genes involved in symbiotic signals, such as NF or Nops production, upon calcium, apigenin, and both treatments together. We detected that Nops production is drastically reduced in the presence of high amounts of calcium. Surprisingly, we identified that the NopA protein exhibited a cleavage process in the presence of apigenin and calcium, indicating that calcium may have an important role in Nops secretion.

## 2. Results

### 2.1. Calcium Inhibits the Secretion of NGR 234 Nops

*Sinorhizobium fredii* NGR 234, *S. fredii* USDA257, and *S. fredii* HH103 secretes several nodulation outer proteins (Nops) into the extracellular milieu, when grown in the presence of flavonoids in the growth media [19,22,23]. In a previous work, it has been demonstrated that flavonoid-induced *S. fredii* USDA257 cultures in the presence of increasing amounts of calcium-reduced Nops production secreted into the extracellular media [21]. In this work, we decided to investigate whether this reduction also occurs in NGR234. For this purpose, we isolated NGR234 Nops produced in the presence of 1 µM of apigenin and increasing levels of Ca^2+^ and analyzed them by SDS-PAGE and subsequent silver staining (Figure 1). When NGR 234 was grown in the presence of 1 µM apigenin without any added calcium, several Nops including NopX, NopL, NopP, NopB, NopC, and NopA accumulated in the extracellular milieu (Figure 1). Interestingly, the addition of 1 mM Ca^2+^ to the growth medium resulted in a drastic reduction in the Nops accumulation observed when NGR 234 was grown in the presence of 1 mM calcium (Figure 1). A densitometer scan of the stained gel clearly demonstrated the severe reduction of Nops accumulation in culture media containing 1 mM Ca^2+^ (Appendix A).

### 2.2. Calcium Promotes the Cleavage of NopA

Calcium at higher concentration (1 mM) inhibited the secretion of NGR234 Nops into the extracellular media (Figure 1). Interestingly, two new proteins were also detected in the extracellular media when NGR234 was grown in the presence of apigenin and 0.5–1 mM calcium (Figure 1 and Figure 2). The new proteins appeared to have low molecular weight and were not seen in the culture supernatants when calcium concentration was lower than 0.5 mM (Figure 1). We wanted to examine if these low-molecular-weight proteins represent cleaved products of NopA or new calcium-induced proteins. To test our hypothesis, we performed immunoblot analysis using antibodies specific for NopA. The NopA antibodies recognized the 8 kDa protein when NGR234 was grown in the presence of 1 µM apigenin (Figure 2A, labeled ‘A’). In addition to this 8 kDa protein, one additional low-molecular-weight protein (Figure 2A, labeled ‘B’) was also recognized by NopA antibodies when NGR234 was grown in the presence of 1 µM apigenin and 0.5 mM calcium (Figure 2A). However, the NopA antibodies failed to recognize the lower molecular weight protein (Figure 2A, labeled ‘C’) indicating that either this peptide does not contain the epitope recognized by the NopA antibody or it could be a new calcium-induced protein. To verify the identity of these low-molecular-weight proteins, we excised gel slices corresponding to NopA (Band A) and the two other peptides (Bands B and C) and analyzed them by mass spectrometry. This analysis revealed that band A, B, and C showed significant sequence homology to the *S. fredii* NGR234 NopA (Appendix A). Several peptides from band A, B, and C gave statistically significant protein scores for the matches with NopA, with MOWSE scores above the 95% confidence level (Appendix A). Band A was composed of peptides that covered the full length of NopA while the peptides included in band B and C matched 85% and 87% amino acid sequences of NopA. Our analysis confirmed that band B and C both correspond to NopA fragments although band C was not recognized by the NopA antibody (Figure 2).

### 2.3. Global Gene Expression Analysis

In order to analyze the impact of the addition of calcium and apigenin on the global gene expression of NGR234, we performed RNA-seq experiments. We compared the transcriptome profile at the early stationary growth phase of NGR234 cultures that were grown in YEM under the following conditions: (1) in the presence of calcium at 0.5 mM concentration, (2) apigenin at 1 µM, (3) 0.5 mM calcium and 1 µM apigenin, and (4) control condition (no calcium or apigenin added). For each condition, three independent biological sample experiments were performed, and their corresponding RNA samples were obtained. Thus, 12 cDNA libraries were generated and sequenced, obtaining between 39 and 86 million reads in each condition. In general, all the samples showed about 99.6% alignment when they were mapped to the NGR234 genome (Genome assembly GCF_000018545.1).

The differentially expressed genes (DEGs) were considered as statistically significant when the fold change ratio between two conditions was ≥3 and had an adjusted *p* value of ≤0.05. The DEGs of each comparison are listed in the Appendix A, where the shown data are the mean of the three replicates in each condition. The transcriptome of NGR234 in the presence of apigenin, calcium, and both compounds together showed a total of 150 (131 upregulated and 19 downregulated), 65 (39 upregulated and 26 downregulated), and 187 (162 upregulated and 25 downregulated) DEGs, respectively, compared to the NGR234 grown in control conditions (Table 1). 

The NGR234 genome is composed of 6443 genes; the supplementation of apigenin, calcium and the combination of both compounds affected the expression of about 2.3%, 1.0%, and 2.9% of the genome of this strain, respectively. Depending on the condition, the number of DEGs in each replicon is different. For example, the presence of apigenin showed 46.7% out of the total number of DEGs in the symbiotic plasmid (plasmid a), while 56.9% of the DEGs observed in the presence of calcium are located on the chromosome. In the case of the DEGs obtained by the presence of both compounds together, apigenin and calcium, about 81% out of the total number of DEGs are located on the chromosome and plasmid a (Table 2). Curiously, 87.3% of the DEGs seen in the presence of apigenin were upregulated (independently of the replicon analyzed), while calcium treatment upregulated and repressed 60% and 40% of the DEGs, respectively.

### 2.4. Effect of Apigenin on the NGR234 Transcriptome

Previous transcriptomic studies carried out in the presence of inducer flavonoids demonstrated a profound effect on the global gene expression in several rhizobia and have identified numerous differentially expressed genes [18]. Most of the genes that are upregulated upon flavonoid treatment were located on the symbiotic plasmids while a limited number of genes of this plasmid were also down regulated [24,25]. In accordance with previous reports, we also found that apigenin regulated the expression of 150 genes, out of which 131 were upregulated and 19 were downregulated (Table 1). The upregulated genes included several nod genes involved in NF production (*nodABC*, *nodS*, *nodU*, *nodZ*), genes involved in the type III secretion system (*rhcJ*, *nolU*, *rhcQ*, *rhcR*, *rhcS*, *rhcT*, *rhcV*, *rhcU*), and genes encoding Nops (*nopA*, *nopB*, *nopL*, *nopP*, *nopT*, *nopC*, *nopM*). These DEGs can be grouped into different classes, namely, genes controlled by a nod box (NB), tts box (TB), and genes that do not contain either NBs or TBs [25]. In the case of *Sinorhizobium fredii* NGR234, 19 NBs (NB1 to NB19) have been identified; 18 out of these 19 NBs were inducible by flavonoid in a NodD1-dependent manner. The induction of four NB-containing genes was found to be dependent on NodD2 [26,27]. Our transcriptome analysis of NGR234 also clearly showed that several nod genes were highly induced by the addition of apigenin (Appendix A). We also analyzed the genes whose expression is driven by the 19 NBs described in NGR234 [28] and the 7 TBs located on the symbiotic plasmid. Clearly, the addition of apigenin upregulated the expression of all the 19 NB- and 7 TB-containing genes (Table 3 and Table 4).

### 2.5. Calcium Represses the Expression of nodABC Genes 

Transcriptome profiling of NGR234 upon calcium treatment revealed 65 differentially expressed genes (Table 1). These DEGs can be assigned into several functional categories based on the KEGG database (http://www.genome.jp/kegg/pathway.html; accessed on 29 May 2023) along with some insertion sequences and hypothetical proteins without known function. Regarding the 39 upregulated genes, two main groups can be extracted based on the processes that they are involved. A set of 12 DEGs was related to amino acid transport and metabolism, and another 12 DEGs were involved in carbohydrate transport and metabolism (Table 5). Calcium had a pronounced effect on the transcription of NGR_b22780, which exhibits a 17-fold change when compared to NGR234 cells grown in the absence of calcium. This DEG encodes a putative transmembrane protein consisting of 88 amino acids. This small protein contains a conserved DUF3311 domain and belongs to a family of short bacterial proteins of unknown function. Calcium also upregulated the expression of several sugar transporters, permeases, aminotransferases, and oxidoreductases (Table 5).

The 26 downregulated genes in the presence of calcium carry out diverse putative functions. One of them, the NGR_a02570 gene, codes for OmpW, an outer membrane protein similar to Omp22. Consistent with our observation, it was earlier shown that the expression of Omp22 under high levels of calcium was reduced [29]. Surprisingly, we found that *nodABC*s, which are involved in NF production in response to plant root flavonoid induction, were included into the repressed genes (−4.53, −4.41, and −3.62, respectively) (Table 4). Additionally, calcium also repressed the expression of several proteins including a cytochrome c-type biogenesis protein, a cytochrome bd ubiquinol oxidase subunit I, and a cytochrome bd ubiquinol oxidase subunit II (Appendix A).

### 2.6. Effect of Calcium plus Apigenin on the NGR234 Transcriptome

Comparative analysis of the DEGs among the calcium, apigenin, and calcium-plus-apigenin treatments with respect to the control conditions showed 16 shared genes, while 42 were uniquely regulated in the presence of apigenin and 16 specifically expressed in the presence of calcium (Appendix A). Surprisingly, the simultaneous presence of both compounds differentially expressed 45 genes which were not shared when either calcium or apigenin were added to the culture. The higher amount of shared DEGs (92) was found between the apigenin and calcium-plus-apigenin conditions (Figure 3). We carried out another interesting analysis by comparing the RNA-Seq data between samples treated with calcium plus apigenin with respect to apigenin (Appendix A). Our analysis showed that most of the DEGs were related to the sugar metabolism and transport (tripartite ATP-independent periplasmic transporters), as well as genes related to the glycerol metabolism. 

Analyzing the shared and specific up- or downregulated genes in more detail, the results showed that the calcium treatment and the combination of calcium plus apigenin shared more percentage of downregulated genes than upregulated genes with respect to the global comparison. On the contrary, the presence of apigenin mainly revealed upregulated shared genes with the calcium-plus-apigenin treatment (Appendix A). As mentioned previously, we observed that calcium treatment repressed the expression of the *nodABC* genes. This observation prompted us to check if other NB- or TB-containing genes were also affected by the calcium addition or the combination of both compounds. We analyzed the genes whose expression is driven by the 19 NBs described in NGR234 [28] and the 7 TBs located on the symbiotic plasmid. Unlike the *nodABC* repression by the presence of calcium, we did not find changes in the expression of genes included in the *nod* regulon or the genes under TB control (Table 3). In addition, the combination of the two conditions did not substantially modify the expression of those genes, only the nolO, y4hM, and fixBC genes increased their expression and are therefore included as DEGs (Appendix A).

### 2.7. Quantitative RT-PCR Analysis Verification of RNAseq Transcriptome Data

To validate the RNA-Seq results, we randomly selected 16 DEGs and examined their transcriptional profile by performing qRT-PCR (Figure 4). Primers that are specific for the amplification of conserved regions of *nodA*, *nodZ*, *rhcJ*, *syrM2*, *nopA*, *tts1*, transcriptional regulators (NGR_b17530; NGR_b03240), sugar ABC transporter, putative outer membrane protein, glycerol-3-phosphate dehydrogenase, aspartate racemase, cytochrome c-type biogenesis protein, dihydrolipoamide dehydrogenase, and succinoglycan biosynthesis protein exoA were synthesized (Appendix A) and were used for performing qRT-PCR analysis (Figure 4). Our results are in agreement with the RNA-seq data. In most cases we found a linear correlation between the fold change values from the RNA-seq and qPCR experiments. Even though the fold changes were not identical, both methods yielded similar expression trends under different treatments.

## 3. Discussion

In the symbiotic process, several molecular signals must be exchanged between the host plant and the bacterial symbiont in a coordinated way. Among these signals, the inducer flavonoids, together with the bacterial regulator NodD, play a critical role since they trigger the expression of genes under the control of NB sequences [6,22]. Among the upregulated genes are those related to NF production as well as several regulators such as ttsI, whose encoded product activates the expression of genes involved in the T3SS, both structural and Nop-coding genes [13,14]. Both NF and Nops act as molecular determinants for the nodulation specificity between the host plant and the rhizobium. Therefore, they are involved in the nodulation host range [5,30,31]. However, little is known about the impact of environmental conditions, physical or chemical, on the production of rhizobial symbiotic signals. One of these elements is calcium availability, which may vary depending on the type of soil. Calcium is a crucial element in the symbiotic process, since it is involved in signal transmission through its concentration oscillations, commonly known as “calcium spiking”, in root cells [8,9,32]. The calcium spiking not only occurs during nodule organogenesis but also in rhizobial cells in response to the detection of plant flavonoids, which allows the expression of nod genes since their transcription is calcium dependent [33].

In this work, we have studied the role of calcium on the global gene expression of NGR234 in the absence and presence of the inducer flavonoid apigenin, and its effect on T3SS function. Our results indicate that calcium only affects the expression of 1% of genes within the entire NGR234 genome with respect to the control condition (Appendix A). The DEGs observed in the presence of 0.5 mM of calcium predominantly belong to specific functional groups, including sugar and amino acid metabolism and transport, although there are others involved in protein and transporter production within rhizobial membranes (Appendix A). Regarding genes related to symbiosis, only the genes responsible for NF core production, *nodABC* genes, appeared as downregulated in comparison to the control (Appendix A). As expected, the presence of apigenin induces all genes belonging to the nod regulon (Appendix A), even in the presence of both treatments, where only some genes slightly increase their expression (less than three-fold between the two conditions, calcium versus apigenin plus calcium (Appendix A). These results are in agreement with previous works in *S. fredii* USDA257, where the expression of T3SS-related genes was not affected by calcium treatment in the presence of inducer flavonoids [21]. As reported in *S. fredii* USDA257, we also detected in NGR234 that calcium downregulated the expression of an outer membrane protein, putative OmpW (NGR_a02570) (Table 3). OmpW might be involved in the membrane integrity in USDA257 [29], but in the case of pathogenic bacteria, such as *Acinetobacter baumannii*, the increased expression of this protein seems to be involved in the antibiotic resistance mechanism together with OmpA [34]. As mentioned before, NGR234_b22780 is the most upregulated gene in the presence of calcium and codes for a putative small transmembrane protein with an unknown functional domain. In addition, genes flanking the NGR234_22780 (NGR234_b22770 and b22790) are also upregulated upon calcium treatment. These genes encode a predicted monocarboxylic acid permease and a putative protein erfK/srfK precursor. NGR234_b22780 contains two predicted domains, a domain with unknown function (DUF5313) and another domain responsible for sodium/glucose cotransporter. In the case of the NGR234_b2290, it contains a predicted L, D-transpeptidase catalytic domain which can act as an L, D-transpeptidase facilitating an alternative pathway for peptidoglycan cross-linking [35].

SDS-PAGE analysis of the NGR234 extracellular proteins in the presence of apigenin and increasing amounts of calcium revealed a reduction of Nops production at the highest calcium concentration (Figure 1, line 6). Surprisingly, the NopA protein not only reduced its protein content but also appeared in two smaller isoforms. Western blot analysis using an antibody against NopA protein indicated the presence of at least two proteins as NopA, a native isoform and another smaller one (band A and B in Figure 2 panel B). Even though band C was not recognized by the NopA antibody (band C in Figure 2 panel A and B), mass spectrometry analysis of the excised band confirmed that band C is also NopA. These results indicate that calcium does not exert its regulatory function at a transcriptional level but post-transcriptionally. As was commented in previous works, T3SS is regulated by proteases in *Yersinia*, *Salmonella*, and *Pseudomonas* [36,37] under stress conditions. A similar process might be happening in NGR234 in the presence of calcium treatment. However, no protease or peptidase present in the NGR234 genome was found to be upregulated in both calcium and calcium plus apigenin conditions even though they could be activated by the presence of calcium. The novel discovery of calcium-mediated cleavage sites in the NopA protein provides new insights about the reduction of the Nops production in this condition. Since NopA is the main subunit of the T3SS pilus [13], the lack of a native NopA protein could block the secretion of the rest of Nops and therefore it could affect the first steps of the symbiotic establishment. This process might be a mechanism by which the increased concentration of calcium in the symbiosome [38] restricts the T3SS production through the NopA cleavage and disassembles its structure blocking the Nops secretion. On the other hand, the high concentration of calcium would activate the expression of sugar and amino acid transporters that might be necessary for the metabolite exchanges between the host plant and the rhizobium in bacteroid state [3]. 

The concentration of calcium in different soil types varies significantly. This variation primarily depends on the parent material and the extent to which weathering and leaching have shaped soil development. The essential role of calcium in nodulation and nitrogen fixation was documented more than 90 years ago [39]. Calcium availability directly influences the establishment of legume–rhizobia symbiosis. Legumes rely on calcium for proper root development and signaling during nodule formation. Deficiency of calcium has been known to adversely affect the infection and nodulation of several legumes. Calcium deficiency was shown to impair nitrogen fixation while high calcium levels increased the number of nodules [40,41]. Richardson and his associates [42] have demonstrated that high Ca^2+^ increased the amount of *nod* gene-inducing compounds in root exudates. Calcium has also been shown to play a direct role in the formation and growth of the infection thread [43] and in the induction of a normal distribution of nodules on the taproot and lateral roots of soybean [44]. Several studies have demonstrated that the initiation of nodule formation involves a calcium-dependent signal transduction system [8,9,32]. This system triggers metabolic changes that culminate in cell division and nodule development. The effectiveness of nodule formation depends on the optimal concentration of calcium. Too much or too little calcium can disrupt the Nops function and affect symbiosis. Our study demonstrates the important role of calcium in regulating Nops production in NGR234. In summary, Nops are critical players in symbiosis, and the effect of calcium on Nops production may serve as a regulatory mechanism ensuring harmony between rhizobia and leguminous plants in soil. 

## 4. Materials and Methods

### 4.1. Basic Molecular and Microbiological Techniques 

*Sinorhizobium fredii* NGR234 [16] was grown at 30 °C in yeast extract/mannitol (YEM) medium. Apigenin was dissolved in ethanol at a concentration of 1 mg/mL and used at 1 μg/mL. A sterile stock solution of 100 mM of Calcium chloride (Sigma, St. Louis, MO, USA) was prepared and used at various concentrations ranging from 1 µM to 1 mM. All primer pairs used in this work are listed in Appendix A.

### 4.2. Culture Conditions and RNA Extraction 

*Sinorhizobium fredii* NGR234 was grown in YEM media at 30 °C until stationary phase (OD_600_ ≈ 1.2). One mL of stationary phase NGR234 culture was then transferred to 250 mL Erlenmeyer’s flasks, each containing 100 mL of sterile YEM media. Nops production was triggered by the addition of 1 µM apigenin to the culture media. The effect of calcium on Nops production was studied by the addition of 500 µM calcium to YEM media that was also supplemented with 1 µM apigenin. 

Total RNA was isolated using the High Pure RNA Isolation Kit (Thermo Scientific, Waltham, MA, USA) according to the manufacturer’s instructions. Verification of the amount and quality of the resulting total RNA samples was carried out by using a Nanodrop 1000 spectrophotometer (Thermo Scientific, Waltham, MA, USA) and a Qubit 2.0 Fluorometer (Invitrogen, Waltham, MA, USA). Three independent total RNA extractions were obtained for each condition.

### 4.3. RNA Isolation and Sequencing

Ribosomal RNA was depleted using a MICROB Express Bacterial mRNA Purification kit (Ambion, Austin, TX, USA), following the manufacturer’s protocol. The integrity and quality of the ribosomal depleted RNA were checked with Agilent Bioanalyzer 2100 (Agilent Technologies, Santa Clara, CA, USA). RNA sequencing was carried out by a company with the Next Generation Sequence (NGS) platform Illumina using the Illumina HiSeq 2000 sequencing instrument (Illumina, San Diego, CA, USA). Ribosomal-depleted samples were used to generate whole transcriptome libraries following the manufacturer’s recommendations for sequencing on this NGS platform. Amplified cDNA quality was analyzed by the Bioanalyzer 2100 DNA 1000 kit (Agilent Technologies) and quantified using the Qubit 2.0 Fluorometer (Invitrogen). 

#### 4.3.1. Mapping of the RNA-Seq Data 

The initial whole transcriptome paired-end reads obtained from sequencing were mapped against the latest version of the *S. fredii* NGR234 genome (https://www.ncbi.nlm.nih.gov/assembly/GCF_000018545.1/; accessed on 1 December 2022) using the Life Technologies mapping algorithm version 1.3 (http://www.lifetechnologies.com/).

#### 4.3.2. Assessment of Differentially Expressed Genes 

The gene expression levels were calculated using the Bioconductor Packages (DESeq2) version 1.36.0 [45], Rbowtie2 version 2.2.0 [46], Rsamtools version 2.2.3 [47], and Rsubread version 2.10.4 [48], bedtools version 2.30.0 [49], and samtools version 1.15.1 [50] software. Differentially expressed genes were defined as those genes with a fold-change lower or higher than −3 or 3, respectively, with a *p* value lower than 0.05.

#### 4.3.3. General Features of the Total Sequenced and Mapped Reads 

Reads were mapped and paired using the software mentioned above. Three biological and independent experiments were carried out for each condition (Appendix A).

#### 4.3.4. RNA-Seq Data Accession Number

The RNA-seq data discussed in this publication have been deposited in the Sequence Read Archive of NCBI (BioProject database) under the BioProject ID PRJNA1024659.

### 4.4. Quantitative Reverse Transcription PCR 

Results obtained in the RNA-Seq analysis were validated by quantitative reverse transcription PCR (qRT-PCR) of 16 selected genes, which represented differentially and non-differentially expressed genes in the four conditions (control, Apigenin, calcium, apigenin + calcium). Total RNA was isolated using the High Pure RNA Isolation kit (Roche, Basel, Switzerland) and RNase Free DNA Set (Qiagen, Chuo City, Tokyo) according to the manufacturer’s instructions. This (DNA free) RNA was reverse transcribed to cDNA by using PrimeScript RT reagent kit with gDNA Eraser (Takara, San Jose, CA, USA). Quantitative PCR was performed using a LightCycler 480 (Roche, Switzerland) with the following conditions: 95 °C, 10 min; 95 °C, 30 s; 50 °C, 30 s; 72 °C, 20 s; forty cycles, followed by the melting curve profile from 60 to 95 °C to verify the specificity of the reaction. The S. fredii NGR234 16S rRNA was used as an internal control to normalize gene expression. The selected genes and primers are listed in Appendix A.

### 4.5. Purification and Analysis of Nops 

Nodulation outer proteins (Nops) were isolated as described earlier [22]. Nops were separated by SDS-PAGE using the discontinuous buffer system of Laemmli [51]. An equal volume of protein was loaded in each lane taking into consideration that protein extractions were carried out from the same volume of cultures at the same growth stage with similar cell number. Electrophoresis was performed on 15%(*w*/*v*) SDS polyacrylamide gels. After the electrophoresis, proteins were stained with Coomassie Blue. 

### 4.6. Western Blot Analysis

Nodulation outer proteins (Nops) of NGR234 that were grown in the presence or absence of apigenin and calcium were separated by 1D SDS-PAGE. Resolved proteins were electrophoretically transferred to a nitrocellulose membrane. The membrane was incubated with polyclonal antibodies raised against S. fredii USDA257 NopA protein that was diluted 1:10,000 in Tris-buffered saline (TBS; 10 mM Tris-HCl [pH 7.5], 500 mM NaCl) containing 5% nonfat dry milk. Following overnight incubation, the membrane was washed three times with TBST (TBS containing 0.3% Tween 20). The membrane was then incubated with goat anti-rabbit IgG–horseradish peroxidase conjugate which was diluted 1:10,000 in TBST containing 5% nonfat dry milk. After several rinses in TBST, immunoreactive polypeptides were detected with an enhanced chemiluminescent substrate (Super Signal West Pico kit; Pierce Biotechnology, Rockford, IL, USA) according to the manufacturer’s instructions.

### 4.7. Mass Spectrometry Analysis

Protein bands corresponding to NopA and the processed peptides were excised from the acrylamide gel, washed in distilled water, and then destained in a 50% solution of acetonitrile (*v*/*v*) containing 25 mM of ammonium bicarbonate. After destaining, a 100% acetonitrile wash was performed and digested with 20 μL (10 μg/mL) of modified porcine trypsin in 25 mM ammonium bicarbonate (Promega, Madison, WI, USA). The peptides resulting from tryptic digestion were then analyzed by liquid chromatography mass spectrometry (LCMS). Database searches were conducted using Proteome discoverer. 

## Figures and Tables

**Figure 1 ijms-25-03443-f001:**
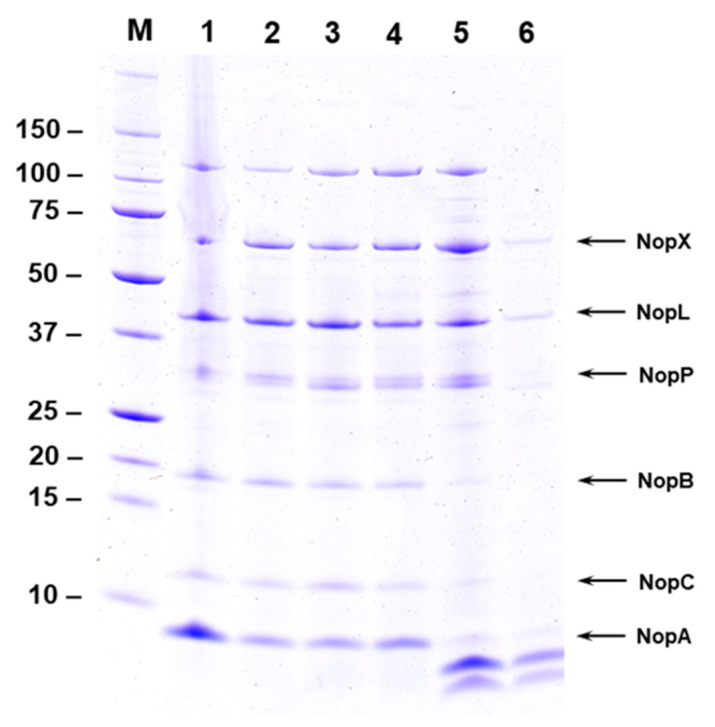
Effect of calcium on Nops secretion. Nops from *S. fredii* NGR234 grown for 48 h in presence of 1 µM apigenin and increasing concentration of calcium were isolated and separated by SDS-PAGE. Resolved proteins were visualized by silver stain. NGR234 was cultured in YEM medium containing 1 µM apigenin and 1 µM (lane 2), 10 µM (lane 3), 100 µM (lane 4), 500 µM (lane 5), and 1000 µM (lane 6) of calcium. Nops isolated from NGR234 grown in the presence of 1 µM apigenin without any added calcium is shown in lane 1. The identity of Nops and the sizes of molecular weight markers are indicated on the sides of the figure.

**Figure 2 ijms-25-03443-f002:**
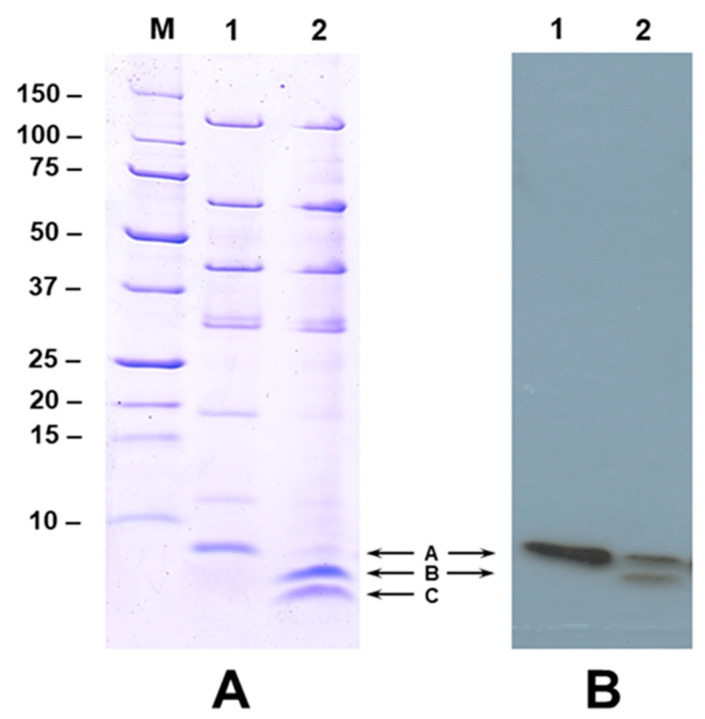
Calcium promotes the cleavage of NopA. Panel **A**. SDS-PAGE analysis of NGR234 Nops produced in the presence of 1 µM apigenin (lane 1) and 1 µM apigenin and 500 µM of calcium (lane 2). Nops were separated on a 15% acrylamide gel and visualized by Coomassie Blue stain. Panel **B**. Immunoblot analysis of NopA cleavage. Proteins shown in Panel A were transferred to nitrocellulose membrane and incubated with NopA specific polyclonal antibodies. The immunoreactive polypeptides were detected by the chemiluminescent detection method. The position of NopA and cleaved NopA products are shown with arrows.

**Figure 3 ijms-25-03443-f003:**
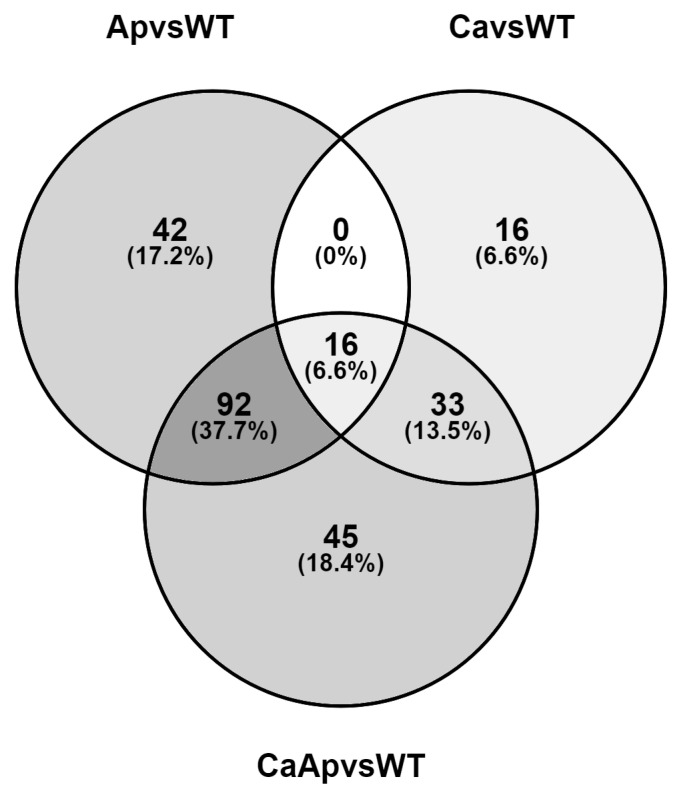
Venn diagram showing the overlap of differentially expressed genes among the calcium, apigenin, and calcium-plus-apigenin treatments with respect to the control condition. The diagram shows the number of uniquely regulated genes and the number of commonly regulated genes in each treatment compared to the control condition.

**Figure 4 ijms-25-03443-f004:**
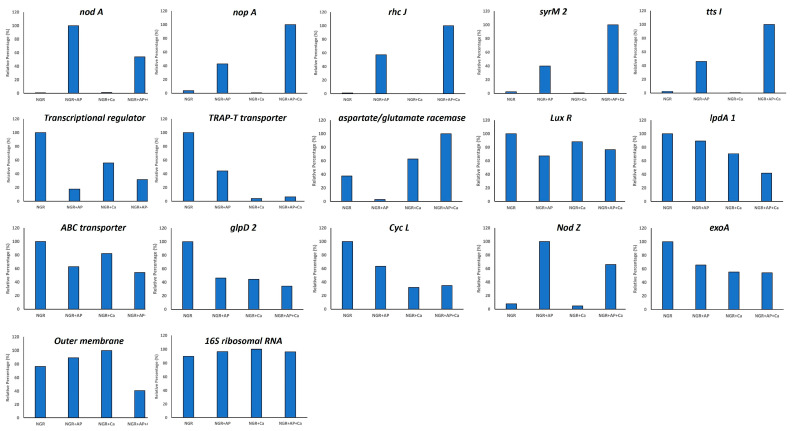
qRT-PCR validation of select differentially expressed genes identified by RNAseq analysis.

**Table 1 ijms-25-03443-t001:** Summary of *S. fredii* NGR234 DEGs in each condition and replicon. WT, control conditions; AP, apigenin; Ca, calcium.

Replicon/Condition		APvsWT	CavsWT	CaAPvsWT
chromosome	Upregulated	39	19	58
	Downregulated	7	18	18
plasmid a	Upregulated	67	0	74
	Downregulated	3	5	3
plasmid b	Upregulated	25	20	30
	Downregulated	9	3	4
	Upregulated	131	39	162
	Downregulated	19	26	25
	Total	150	65	187

**Table 2 ijms-25-03443-t002:** Summary of DEGs percentage of *S. fredii* NGR234 in each condition and replicon. WT, control conditions; AP, apigenin; Ca, calcium.

Replicon/Condition	APvsWT	CavsWT		CaAPvsWT
chromosome	30.7	56.9		40.6
plasmid a	46.7	7.7		41.2
plasmid b	22.7	35.4		18.2
Replicon/Condition	Upregulated		Downregulated	
APvsWT	87.3		12.7	
CavsWT a	60.0		40.0	
CaAPvsWT	86.6		13.4	

**Table 3 ijms-25-03443-t003:** Induction on NGR234 ORFs regulated by apigenin, calcium, and both treatments located downstream of *nod* boxes.

Locus-Tag	Annotation	NB ^a^	Ap Induction ^b^	Ca Induction ^b^	Ap + Ca Induction ^b^	Ap/Ap + Ca	Description
NGR_a00430	*y4aD*	1	23.64	0.83	26.89	0.88	putative phytoene synthase-like protein
NGR_a00440	*y4aC* (*hpnD*)	1	11.90	0.83	13.20	0.90	squalene synthase HpnD
NGR_a00450	*y4aB*	1	16.84	1.07	19.11	0.88	putative flavoprotein oxidoreductase
NGR_a00460	*y4aA* (*shc*)	1	2.39	0.82	2.82	0.85	probable squalene-hopene cyclase
NGR_a00400	*nodZ*	2	2.17	0.35	2.62	0.83	chitin oligosaccharide fucosyltransferase nodulation protein NodZ
NGR_a00410	*noeL*	2	1.92	0.52	2.08	0.93	GDP-mannose 42C62C dehydratase nodulation protein NoeL
NGR_a00420	*nolK*	2	1.34	0.66	1.45	0.92	nucleotide sugar epimerase/dehydrogenase nodulation protein NolK
NGR_a00360	*nodD1*	3	0.35	0.90	0.26	1.35	LysR-type transcription regulator of nodulation genes nodulation protein NodD1
NGR_a03860	*nolL*	4	6.02	0.73	7.86	0.77	nodulation protein NolL involved in acetylation of Nodulation factor
NGR_a03530	*fixF*	6	18.66	0.97	29.50	0.63	involved in polysaccharide synthesis/modification
NGR_a03480	*noeE*	7	9.36	0.43	9.95	0.94	nodulation protein NoeE involved in sulfation of Nodulation factor
NGR_a03410	*nodA*	8	5.32	0.22	6.56	0.81	N-acyltransferase nodulation protein NodA involved in Nodulation factor synthesis
NGR_a03420	*nodB*	8	4.32	0.23	5.03	0.86	chitooligosaccharide deacetylase nodulation protein NodB involved in Nodulation factor synthesis
NGR_a03430	*nodC*	8	3.90	0.28	4.32	0.90	N-acetylglucosaminyltransferase nodulation protein NodC involved in Nodulation factor synthesis
NGR_a03440	*nodI*	8	4.58	0.50	5.91	0.77	ABC-transporter ATP-binding protein NodI involved in secretion of Nodulation factor
NGR_a03450	*nodJ*	8	3.92	0.57	5.33	0.74	ABC-transporter permease NodJ probably involved in secretion of Nodulation factor
NGR_a03460	*nolO*	8	2.52	0.68	3.49	0.72	nodulation protein NolO involved in carbamoylation of Nodulation factor
NGR_a03470	*noeI*	8	1.56	0.81	2.13	0.74	nodulation protein NoeI involved in 2-O methylation of Nodulation factor
NGR_a03370	*y4hM*	9	2.78	0.74	5.12	0.54	conserved hypothetical 43.9 kDa oxidoreductase domain-containing protein
NGR_a03360	*y4hN*	9	4.01	1.07	10.28	0.39	putative transposase number 1 for insertion sequence NGRIS-15b
NGR_a03350	*y4hO*	9	3.89	1.10	4.61	0.85	putative transposase number 2 for insertion sequence NGRIS-15b
NGR_a03310	*y4iR* (similar to PsiB)	10	8.91	0.81	11.11	0.80	conserved hypothetical 17.6 kDa protein
NGR_a02560	*y4mC*	11	1.45	0.77	1.91	0.76	precursor of 26.2 kDa for conserved putative periplasmic protein
NGR_a02390	*nodS*	12	9.26	0.48	6.69	1.38	nodulation protein NodS involved in N-methylation of Nodulation factor
NGR_a02400	*nodU*	12	7.23	0.57	5.71	1.27	nodulation protein NodU involved in C-6 carbamoylation of Nodulation factor
NGR_a02380	*y4nD*	13	6.56	1.01	8.10	0.81	putative transposase number 2 of insertion sequence NGRIS-5b
NGR_a01210	*y4vC*	14	30.87	1.10	52.60	0.59	conserved hypothetical 11.0 kDa protein possibly involved in assembly of iron-sulfur cluster
NGR_a01220	*fixA*	14	3.90	0.89	7.50	0.52	nitrogen fixation protein FixA electron transfer flavoprotein beta chain
NGR_a01230	*fixB*	14	2.93	0.84	5.94	0.49	nitrogen fixation protein FixB electron transfer flavoprotein alpha chain
NGR_a01240	*fixC*	14	2.96	0.81	6.02	0.49	nitrogen fixation protein FixC flavoprotein-ubiquinone oxidoreductase
NGR_a01250	*fixX*	14	1.42	0.78	2.54	0.56	nitrogen fixation ferredoxin-like protein FixX
NGR_a00990	*y4wE*	15	5.08	0.91	11.97	0.42	conserved monooxygenase oxidoreductas of 37.7 kDa protein involved in flavonoid-dependent IAA synthesis
NGR_a01000	*y4wF*	15	15.47	0.91	30.85	0.50	conserved histidinol-phosphate aminotransferase-like protein involved in flavonoid-dependent IAA synthesis
NGR_a00970	*y4wH*	16	63.29	1.13	80.03	0.79	conserved hypothetical 15.6 kDa protein
NGR_a00920	*y4wM*	17	65.17	1.16	91.03	0.72	conserved periplasmic solute-binding protein of ABC-transporter
NGR_a00800	*ttsI*	18	46.99	1.07	61.58	0.76	transcription regulator of late flavonoid-inducible functions TtsI
NGR_a00790	*rhcC2*	18	25.24	0.89	35.68	0.71	outermembrane protein RhcC2 component of type III secretion apparatus
NGR_a00780	*y4xK*	18	19.69	0.81	29.76	0.66	conserved putative lipoprotein of 20.5 kDa
NGR_a00470	*syrM2*	19	16.43	1.04	21.64	0.76	LysR-type transcription regulator SyrM2 involved in flavonoid dependent regulation

^a^ According to [2]. ^b^ Fold induction with respect to non-induced cultures. We considered differentially expressed genes to be those with a fold change of + or − 3. In red, genes upregulated. In blue, genes downregulated. Ap, apigenin; Ca, calcium, NB, *nod* boxes.

**Table 4 ijms-25-03443-t004:** Induction on NGR234 ORFs regulated by apigenin, calcium, and both treatments located downstream of *tts* boxes.

Locus-Tag	Annotation	TB ^a^	Ap Induction ^b^	Ca Induction ^b^	Ap + Ca Induction ^b^	Ap/Ap + Ca	Description
not present		1					
not present		2					
NGR_a03640	*nopM*	3	9.48	0.79	9.35	1.01	nodulation outer protein NopM probable type III effector
not present		4					
not present		5					
NGR_a00700	*nopX*	8	86.00	1.06	88.52	0.97	nodulation outer protein NopX probable subunit of type III translocon;
NGR_a00710	*y4yB*	8	83.49	1.10	87.98	0.95	conserved hypothetical 17.1 kDa protein second copy encoded by pNGR234b
NGR_a00720	*y4yA*	8	52.60	1.13	60.25	0.87	conserved uncharacterized 49.9 kDa enzyme second copy encoded by pNGR234b
NGR_a00730	*y4xP*	8	54.11	1.02	63.29	0.85	conserved putative cysteine synthase second copy encoded by pNGR234b
NGR_a00740	*y4xO*	8	55.91	1.01	66.23	0.84	conserved putative oxidoreductase second copy encoded by pNGR234b
NGR_a00750	*y4xN*	8	55.50	1.17	62.95	0.88	conserved putative siderophore biosynthesis protein second copy encoded by pNGR234b
NGR_a00760	*y4xM*	8	53.93	1.27	57.31	0.94	conserved uncharacterized MFS-type transporter second copy encoded by pNGR234b
NGR_a00770	*nopL*	9	69.17	1.41	66.51	1.04	nodulation outer protein NopL probable type III effector
NGR_a00680	*nopB*	10	140.47	1.39	133.77	1.05	nodulation outer protein NopB precursor of type III secretion pilus subunit
NGR_a00670	*rhcJ*	10	167.54	1.44	151.03	1.11	outermembrane protein RhcJ component of type III secretion apparatus
NGR_a00660	*nolU*	10	167.96	1.22	135.87	1.24	nodulation protein NolU possibly linked to type III secretion
NGR_a00650	*rhcL*	10	135.83	1.00	117.20	1.16	nodulation protein RhcL putative regulator of RhcN activity in type III secretion
NGR_a00640	*rhcN*	10	110.11	0.99	100.64	1.09	ATPase RhcN for type III secretion
NGR_a00630	*y4yJ*	10	67.66	1.00	61.69	1.10	conserved hypothetical 20.4 kDa protein possibly linked to type III secretion
NGR_a00620	*rhcQ*	10	60.21	1.04	59.86	1.01	component RhcQ of type III secretion apparatus
NGR_a00610	*rhcR*	10	49.42	1.12	58.60	0.84	innermembrane protein RhcR component of type III secretion apparatus
NGR_a00600	*rhcS*	10	44.43	1.32	54.23	0.82	innermembrane protein RhcS component of type III secretion apparatus
NGR_a00590	*rhcT*	10	29.69	1.14	34.79	0.85	innermembrane protein RhcT component of type III secretion apparatus
NGR_a00580	*rhcU*	10	4.63	0.76	4.48	1.03	innermembrane protein RhcU component of type III secretion apparatus
NGR_a00570	*nopP*	11	80.67	1.29	76.14	1.06	nodulation outer protein NopP probable type III effector
NGR_a00560	*nopC*	12	12.05	1.05	11.66	1.03	nodulation outer protein NopC linked to type III secretion pilus
NGR_a00550	*nopA*	12	14.07	1.03	14.14	0.99	nodulation outer protein NopA precursor of type III secretion pilus subunit
NGR_a00540	*y4yQ*	12	14.17	1.00	14.60	0.97	conserved hypothetical 31.2 kDa membrane protein possibly linked to type III secretion
NGR_a00530	*rhcV*	12	26.89	0.94	34.03	0.79	Inner membrane protein RhcV component of type III secretion apparatus
NGR_a00520	*y4yS*	12	19.06	0.94	25.73	0.74	conserved hypothetical 20.1 kDa TPR repeat-containing protein possibly linked to type III secretion
NGR_a00490	*nopT*	13	22.72	0.96	20.07	1.13	nodulation outer protein NopT probable type III effector
NGR_a00480	*y4zD*	13	4.88	0.99	4.35	1.12	hypothetical 5.5 kDa protein

^a^ According to [12]. ^b^ Fold induction with respect to non-induced cultures. We considered differentially expressed genes those with a fold change of + or − 3. In red, genes upregulated. In blue, genes downregulated. Ap, apigenin; Ca, calcium, TB, *tts* boxes.

**Table 5 ijms-25-03443-t005:** Functional characterization on the NGR234 DEGs upon calcium treatment compared to the control conditions.

Gene_Name	FoldChange	Description
Amino acid metabolism		
aatA1	3.360571931	Aspartate aminotransferase B
NGR_b03050	3.21494603	putative acetolactate synthase II large subunit
NGR_b03060	3.235062489	putative FAD dependent oxidoreductase
NGR_b03070	3.349198302	dTDP-glucose 4,6-dehydratase
NGR_b03090	3.186166763	N-methylhydantoinase B
NGR_b22770	3.239904207	predicted monocarboxylic acid permease
NGR_c00380	0.324639305	putative peptidase M22 glycoprotease
NGR_c00390	0.316690296	predicted ribosomal-protein-alanine N-acetyltransferase
NGR_c25920	4.635364648	homoisocitrate dehydrogenase
NGR_c26220	3.660494863	Serine—glyoxylate aminotransferase
NGR_c26230	5.687498036	putative citrate lyase
putA2	6.255907925	bifunctional PutA protein
Energy production and conversion		
cycH	0.2836311	cytochrome c-type biogenesis protein CycH
cycK	0.260344537	cytochrome c-type biogenesis protein
cycL	0.228701936	cytochrome c-type biogenesis protein CycL precursor
NGR_c07800	0.205481298	putative cytochrome c-type biogenesis protein
Sugar metabolism and trasport		
dctP	5.291970719	TRAP dicarboxylate transporter-DctP subunit
dctQ	3.03588033	tripartite ATP-independent periplasmictransporter DctQ component
NGR_b22590	3.349646746	putative C4-dicarboxylate transport system permease protein
NGR_c14470	0.241135103	putative malonate transporter
NGR_c14480	0.264455112	2-octaprenyl-6-methoxyphenol hydroxylase
NGR_c16750	3.270277309	putative ATP-binding component of ABC transporter
NGR_c17820	3.479463771	putative transmembrane component of ABC transporter
NGR_c17830	5.767996243	probable sugar ABC transporter substrate-binding protein
NGR_c30950	4.271123502	putative ATP-binding protein of sugar ABC transporter
NGR_c30960	3.952674083	putative ATP-binding protein of sugar ABC transporter
NGR_c30970	4.03463148	putative permease component of ABC transporter
NGR_c30980	3.840488791	putative permease component of ABC transporter
NGR_c30990	3.026543595	putative transmembrane protein
thuE	3.61249158	ABC transporter sugar-binding protein
Energy production and conversion/Protein turnover, chaperones		
glpD2	5.238406536	glycerol-3-phosphate dehydrogenase
glpR	4.505200318	glycerol-3-phosphate transcriptional regulator protein DeoR family
NGR_c01900	0.079540905	cytochrome bd ubiquinol oxidase subunit II
NGR_c01910	0.088986153	cytochrome bd ubiquinol oxidase subunit I
senC	0.328098885	putative electron transport protein
Cell wall/membrane/envelope biogenesis]		
NGR_b22790	4.580477143	Protein erfK/srfK precursor (Lipoprotein-anchoring transpeptidase ErfK/SrfK)
NGR_a02570	0.254946475	conserved putative outer membrane protein of 24.6 kDa (OmpW)
Periplasmic proteins		
NGR_b05040	3.301240631	putative periplasmic protein
NGR_b00380	0.297100625	putative periplasmic protein
Metal ion binding		
NGR_b05000	3.064244497	conserved hypothetical protein (HupE/ureJ) nickel binding
NGR_b01360	3.084046646	hypothetical protein
Transmembrane proteins		
NGR_c19900	0.319606552	putative transmembrane protein
NGR_b22780	17.58100064	putative transmembrane protein
NFs production		
nodA	0.220941954	N-acyltransferase nodulation protein NodA involved in Nodulation factor synthesis
nodB	0.226676991	chitooligosaccharide deacetylase nodulation protein NodB involved in Nodulation factor synthesis
nodC	0.276039723	N-acetylglucosaminyltransferase nodulation protein NodC involved in Nodulation factor synthesis
Hypothetical proteins		
NGR_b03420	3.717502659	hypothetical protein
NGR_b04970	3.22988419	conserved hypothetical protein
NGR_b04980	3.561955755	conserved hypothetical protein
NGR_b09710	3.202250736	hypothetical protein
NGR_c03830	4.686157099	hypothetical protein
NGR_c03900	0.265386827	hypothetical protein
NGR_c17840	4.790515014	hypothetical protein
NGR_c20330	4.411039572	hypothetical protein
NGR_c24510	0.322154807	conserved hypothetical protein
NGR_c31210	0.106635366	hypothetical protein
Other functions		
NGR_a01890	0.252290146	putative transposase of undefined mobile element
NGR_b05010	3.132126719	conserved hypothetical protein (rotamase, protein folding)
NGR_b12570	0.302431844	catalase-peroxidase protein
NGR_b18120	3.402340713	putative cytoplasmic protein
NGR_b18630	3.194922855	alpha proteo sRNA
NGR_b22420	0.221690152	putative transposase number 3 for disrupted insertion sequence NGRIS-6a
NGR_c05880	0.284555583	putative transposase number 3 for insertion sequence NGRIS-11a
NGR_c24500	0.300159215	putative globin family protein
NGR_c25410	0.286628458	aldehyde dehydrogenase

## Data Availability

The RNA-seq data discussed in this publication have been deposited in the Sequence Read Archive of NCBI (BioProject database) under the ID PRJNA1024659. The original contributions presented in the study are included in the article and Appendix A. Further inquiries can be directed to the corresponding author.

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
