# Peer review of "Calcium Induces the Cleavage of NopA and Regulates the Expression of Nodulation Genes and Secretion of T3SS Effectors in Sinorhizobium fredii NGR234"

_ijms, 2024, doi:10.3390/ijms25063443_

Round 1

Reviewer 1 Report

Comments and Suggestions for Authors

The authors performed extraction of Nops secreted proteins as well as transcriptome analysis under Ca treatment and identified a number of differentially expressed genes in NGR234. The results of this manuscript provide new insights into the role of calcium in rhizobia-legume interactions. However, there are some problems here, and it is suggested that the manuscript needs major revision.

1.     For Figure 1, it is suggested that a control of NGR234ΩttsI or rhcN mutant secreted by Nops should be added to further confirm that this result is not an effect of rhizobia cell rupture of the content proteins on this experiment.

2.     The title of the authors' manuscript is “Calcium induces the cleavage of NopA and regulates the expression and secretion of T3SS effectors in Sinorhizobium fredii NGR234”, but in the results, the authors emphatically describe the expression of nodulation genes and OmpW in the transcriptome, which is somewhat at odds with the title. It is suggested that the authors revise the title of the manuscript.

3.     Suggestion to add significance analysis in figure 4

Some minor problems:

4.     Line 341; SDS page should be SDS-PAGE.

5.     Line 101; Ca2+ should be Ca2+.

6.     Line 136; 8 kDa?

7.     There are a lot of little mistakes in Materials and Methods, such as 1µM (should be 1 µM; Line383), OD600 (should be OD600; Line387) and 250 ml (should be 250 mL; Line388). Authors need to proofread in the revised manuscript.

8.     Line 212; TB; nod boxes should be NB; nod boxes. This is rightAnd in Line 217 TB; nod boxes is tts box.

9.     In Table 4; There are a lot of % in description, such as dTDP-glucose 4%2C6-dehydratase.

Author Response

Response to Reviewer 1

Response shown in red font.

Comments and Suggestions for Authors

The authors performed extraction of Nops secreted proteins as well as transcriptome analysis under Ca treatment and identified a number of differentially expressed genes in NGR234. The results of this manuscript provide new insights into the role of calcium in rhizobia-legume interactions. However, there are some problems here, and it is suggested that the manuscript needs major revision.

  1. For Figure 1, it is suggested that a control of NGR234ΩttsIor rhcN mutant secreted by Nops should be added to further confirm that this result is not an effect of rhizobia cell rupture of the content proteins on this experiment.

Previous studies have shown that NGR234ΩttsI or rhcN mutants cannot secrete Nops [1, 2]. Moreover, we have demonstrated that calcium does not appear to interfere with the secretion of Nops since Western blot analysis utilizing antibodies raised against several Nops revealed that these proteins do not accumulate inside the cell [3].

[1]. Marie et al. 2003. Characterization of Nops, Nodulation Outer Proteins, Secreted Via the Type III Secretion System of NGR234. Molecular plant-microbe interactions : MPMI. 16. 743-5.

[2]. Marie et al.  TtsI, a key regulator of Rhizobium species NGR234 is required for type III-dependent protein secretion and synthesis of rhamnose-rich polysaccharides. Mol Plant Microbe Interact. 2004 Sep;17(9):958-66

[3]. Krishnan et al. 2007. Calcium regulates the production of nodulation outer proteins (Nops) and precludes pili formation by Sinorhizobium fredii USDA257, a soybean symbiont. FEMS microbiology letters.  271. 59-64).

  1. The title of the authors' manuscript is “Calcium induces the cleavage of NopA and regulates the expression and secretion of T3SS effectors in Sinorhizobium frediiNGR234”, but in the results, the authors emphatically describe the expression of nodulation genes and OmpW in the transcriptome, which is somewhat at odds with the title. It is suggested that the authors revise the title of the manuscript.

We have changed the title based on the reviewer’s suggestion.

  1. Suggestion to add significance analysis in figure 4

The results reported are the average of duplicate samples and  hence we are unable to perform significance analysis.

Some minor problems:

  1. Line 341; SDS page should be SDS-PAGE.

Changed as suggested.

  1. Line 101; Ca2+ should be Ca2+.

Changed as suggested.

  1. Line 136; 8 kDa?

Changed as suggested.

  1. There are a lot of little mistakes in Materials and Methods, such as 1µM (should be 1 µM; Line383), OD600 (should be OD600; Line387) and 250 ml (should be 250 mL; Line388). Authors need to proofread in the revised manuscript.

We have made all the corrections pointed out by the reviewer. We have proofread the revised manuscript and made changes when appropriate.

  1. Line 212; TB; nod boxes should be NB; nod boxes. This is right?And in Line 217 TB; nod boxes is tts box.

Changed as suggested.

  1. In Table 4; There are a lot of % in description, such as dTDP-glucose 4%2C6-dehydratase.

We have now rectified this error.

Reviewer 2 Report

Comments and Suggestions for Authors

well written paper focussing on the role of an essential element Ca

few editorial comments noted directly

suggestions for more background details  also shown by sticky notes 

Author Response

We thank the reviewer for his very positive response to our manuscript. We did not find any comments from this reviewer.